# Learning Optical Flow from Continuous Spike Streams

**Rui Zhao**[1,2] **Ruiqin Xiong**[1,2*] **Jing Zhao**[3] **Zhaofei Yu**[1,2,4] **Xiaopeng Fan**[5] **Tiejun Huang**[1,2]

[1]National Engineering Research Center of Visual Technology (NERCVT), Peking University
[2]Institute of Digital Media, School of Computer Science, Peking University
[3]National Computer Network Emergency Response Technical Team
[4]Institute for Artificial Intelligence, Peking University
[5]School of Computer Science and Technology, Harbin Institute of Technology
ruizhao@stu.pku.edu.cn, {rqxiong, yuzf12, tjhuang}@pku.edu.cn,
zhaojing@cert.org.cn, fxp@hit.edu.cn

## Abstract

Spike camera is an emerging bio-inspired vision sensor with ultra-high temporal resolution. It records scenes by accumulating photons and outputting continuous binary spike streams. Optical flow is a key task for spike cameras and their applications. A previous attempt has been made for spike-based optical flow. However, the previous work only focuses on motion between two moments, and it uses graphics-based data for training, whose generalization is limited. In this paper, we propose a tailored network, Spike2Flow that extracts information from binary spikes with temporal-spatial representation based on the differential of spike firing time and spatial information aggregation. The network utilizes continuous motion clues through joint correlation decoding. Besides, a new dataset with real-world scenes is proposed for better generalization. Experimental results show that our approach achieves state-of-the-art performance on existing synthetic datasets and real data captured by spike cameras. The source code and dataset are available at https://github.com/ruizhao26/Spike2Flow.

## 1 Introduction

Neuromorphic camera (NeurCam) is a kind of emerging vision sensor [35; 43; 46; 18; 11; 19] inspired by retina. Using asynchronous sampling models, NeurCams can record the natural scene continuously by firing sparse spike streams with an ultra-high temporal resolution. Besides, NeurCams have high dynamic range, low latency, and low energy consumption. Compared with traditional cameras with single-exposure imaging, NeurCams are more suitable for high-speed imaging in application scenarios such as unmanned aerial vehicles and chemical particle observation. One kind of NeurCams is the event camera [35; 43; 46; 18] inspired by the peripheral retina. It asynchronously fires spikes when the brightness change exceeds a certain threshold in the logarithmic domain. However, event cameras can hardly recover textures due to their differential sampling model, especially for static regions. *Another* kind of emerging NeurCam is the *spike camera* [11; 19] inspired by the fovea of retina. Spike cameras also record the scene by firing spikes asynchronously in each pixel, but different from event cameras that record the relative brightness changes, each pixel in spike cameras encodes the scene by accumulating photons and firing a spike once the accumulation exceeds a certain threshold independently. With such an integral sampling model, spike cameras can record fine textures of the scene, which makes it more appropriate for pixel-level tasks than event cameras. Since spike cameras

---

*Corresponding author.

36th Conference on Neural Information Processing Systems (NeurIPS 2022).

can record high-speed motion and recover the scenes well, they have shown great potential in tasks such as video reconstruction [73; 66; 69; 75; 67; 74], denoising [61], super-resolution [65; 59] and detection [19]. In these tasks, the relative motion between the spike camera and objects in the scene, i.e., the optical flow, is fundamental and pivotal.

Optical flow estimation has been an important and challenging problem since it was proposed [16]. In recent years, there are a deal of developments in optical flow [12; 23; 52; 20; 21; 62; 56; 68; 54; 27; 64]. A simple way to estimate optical flow for spike cameras is reconstructing images and use video-based methods. However, there are two problems. Firstly, simple reconstruction methods may introduce a lot of noise while complex methods have high computational costs [66; 69; 67; 65]. Secondly, simple reconstruction ways such as averaging the spike streams along the temporal axis would lose the precise temporal information in spike streams and introduce blur, which may mislead the flow estimation. Event-based methods [71; 72; 32; 15; 14] also cannot have satisfactory performance for optical flow estimation for spike cameras due to the difference of data modalities. Thus, more efficient methods are needed for estimating optical flow from spike streams. Hu et al. [17] propose SCFlow as an early exploration of this question. They design a pyramidal network and two synthetic datasets based on graphics models to train and evaluate the network, respectively. However, the above-mentioned method is not robust since it omits several issues:

(1) **Efficient representation of binary spikes.** Each spike output by the spike camera represents not the status at the current moment but the result of the integral process of the spike. Besides, a single spike cannot express the information of the corresponding space-time point. Using only convolution to extract features from the spike streams may not be efficient.

(2) **Continuousness of spike streams.** Previous work only considers motion between two single moments in spike streams, and it omits the continuousness information in the moving procedure.

(3) **Reality of the datasets.** Previous work use datasets synthesized by graphics models to train and evaluate the network. However, there is a huge gap between virtual scenes and the real world.

In this paper, we propose the Spike2Flow to estimate flow from continuous spike streams. We propose the differential of spike firing time (DSFT) to transform the binary spikes to better represent the procedure of the integration for each spike. A spatial information aggregation (SIA) module is proposed to aggregate a larger receptive field for each pixel with a self-attention mechanism. The DSFT and SIA form the temporal-spatial representation (TSR) for spike streams. Besides, we propose a joint correlation decoding (JCD) module to use continuous motion clues by simultaneously estimating a series of flow fields. To train and evaluate the network in real scenes, based on scenes in Slow Flow [25], we generate flow fields and spike streams to construct a dataset, i.e., real scenes with spike and flow (RSSF). The main contributions of this paper can be summarized as follows:

(1) A spike-based optical flow network, Spike2Flow, is proposed. The Spike2Flow extracts features from binary spikes with temporal-spatial representation and jointly estimates a series of flow fields to utilize the continuousness of the moving procedure.

(2) A dataset for spike-based optical flow, real scenes with spike and flow (RSSF) is proposed. The scenes in RSSF are from the real world, improving the generalization of networks trained by RSSF.

(3) Experiments demonstrate that the Spike2Flow achieves state-of-the-art performance on RSSF, photo-realistic high-speed motion (PHM) dataset, and real data captured by spike cameras.

## 2    Related Work

**Neuromorphic Cameras.** Neuromorphic cameras (NeurCams) are a kind of vision sensor that gets inspiration from the retina and works asynchronously in each pixel. Event cameras (including DVS [35], DAVIS [43], ATIS [46], CeleX [18] and et al.) and spike cameras [11; 19] are two types of mainstream NeurCams. Both the above-mentioned cameras record the optical scene asynchronously in each pixel, which brings advantages for NeurCams compared with traditional cameras with a single-exposure imaging pattern, such as ultra-high temporal resolution, high dynamic range, low latency, and low energy consumption. Event cameras employ a differential sampling model that only fires events when illuminance change in the logarithmic domain exceeds a certain threshold, while spike cameras use an integral sampling model that accumulates photons and fires spikes when the accumulation exceeds a certain threshold. Thus, the spikes output by event cameras are more

sparse, but fine textures in the scene are lost especially in static regions. More details of the working mechanism of the spike camera can be found in [17; 19; 67; 73; 69; 66].

**Video-Based Optical Flow.** Optical flow estimation aims to find dense pixel correspondences between two moments in videos, which has lots of applications, such as video enhancement [57; 6], frame interpolation [26; 2; 33] and recognition [3; 9; 55]. FlowNet [12] is the first end-to-end flow estimation network trained by the synthetic FlyingChairs dataset. Subsequent works [47; 23; 52; 20; 21; 53; 68] get inspiration from variational methods [4; 51; 49]. They introduce classical knowledge such as the pyramid, coarse-to-fine, and cost volume to the networks. VCN [62] and DICL [56] design new approaches to build more robust correlation between frames. RAFT [54] builds a multi-scale all-pairs cost volume and performs recurrent refinement in a fixed resolution. RAFT has excellent performance and becomes the baseline of the subsequent works [28; 60; 27; 64]. The methods mentioned above are based on supervised learning. There are also some unsupervised optical flow estimation networks [41; 58; 37; 38; 36; 39; 29; 50]. Besides optical flow between two frames, there are also methods for flow among multi-frames. [24; 38; 36] use temporal context by jointly estimating flow from a frame to its previous and future frames. [44; 48; 22] propagate the temporal information from the previous pair of images to the next pair by passing warped flow or latent states of the decoder.

**Event-Based Optical Flow.** There are also several works on event-based optical flow. EV-FlowNet [71] is the first end-to-end deep network, which is trained based on MVSEC [70] dataset. It is in an encoder-decoder fashion and uses gray images from the active pixel sensor (APS) of event cameras to construct the photometric loss. Zhu et al. [72] turn to use the average timestamp of warped events to construct the loss function, leaving the help of gray images. Spike-FlowNet [32] uses both spiking neural networks (SNNs) and analog neural networks (ANNs) to encode the events, achieving lower energy consumption. STEFlow [10] uses recurrent networks to encode the event stream and constructs correlation among features from events through time. Hagenaars et al. [15] estimate event-based flow using deep networks composed fully of SNNs. Gehrig et al. [14] propose E-RAFT to implement all-pairs correlation and recurrent refinement in event-based flow based on a more complex autonomous driving dataset DSEC [13].

## 3 Approaches

### 3.1 Working Mechanism of Spike Cameras

The spike camera is composed of an array of pixels working asynchronously. Each pixel of a spike camera is composed of three main components: photon-receptor, integrator, and comparator. The integrator accumulates the photoelectrons from the photon-receptor and transfers them to the voltage. The comparator compares the accumulation with the threshold continuously. Once the voltage of the integrator exceeds a certain threshold, the camera fires a spike and resets the accumulation. The voltage of the accumulator can be formulated as:

$$A(\mathbf{x}, t) = \int_0^t \alpha \cdot I(\mathbf{x}, \tau) \mathrm{d}\tau \mod \theta \tag{1}$$

where $A(\mathbf{x}, t)$ is the voltage of the accumulator at pixel $\mathbf{x} = (x, y)$. $I(\mathbf{x}, \tau)$ is the lights intensity in pixel $\mathbf{x}$ at time $\tau$. $\theta$ is the threshold of the comparator. The above-mentioned working mechanism is called *Integral-and-Fire* (IF). With such an IF procedure, the spike cameras can fire spikes asynchronously and continuously. The reading time of the spikes is quantified with a period $T$, and the $T$ can reach a micro-second level. The spike camera fires spikes at time $nT, n \in \mathbb{N}$. Thus, the output of the spike camera is a spatial-temporal binary stream $S$ in $H \times W \times N$ size. The $H$ and $W$ are the height and width of the sensor, respectively, and $N$ is the temporal steps of the spike stream.

### 3.2 Problem Statement and Data Generation

**Problem Statement.** Optical flow estimation for spike cameras is to estimate the pixel-level motion field of the projection from the optical scene to the sensor plane. Suppose that we denote a binary spike stream as $S = \{S(\mathbf{x}, t) \mid \mathbf{x} \in \Omega, \ t \in \mathbb{N}, \ t \leq n\}$, if we set $t_{\text{start}} = t_0$ as the starting moment, the optical flow can be denoted as $\mathbf{w} = \{\mathbf{w}(\mathbf{x}, t \mid t_0) \mid t \in \mathbb{N}, \ t_0 \leq t \leq n\}$. The ideally meaning of $\mathbf{w}$ can be formulated as:

$$S(\mathbf{x} + \mathbf{w}(\mathbf{x}, t \mid t_0), \ t) \rightarrow S(\mathbf{x}, t_0), \ \ t \in \mathbb{N}, \ t_0 \leq t \leq n \tag{2}$$

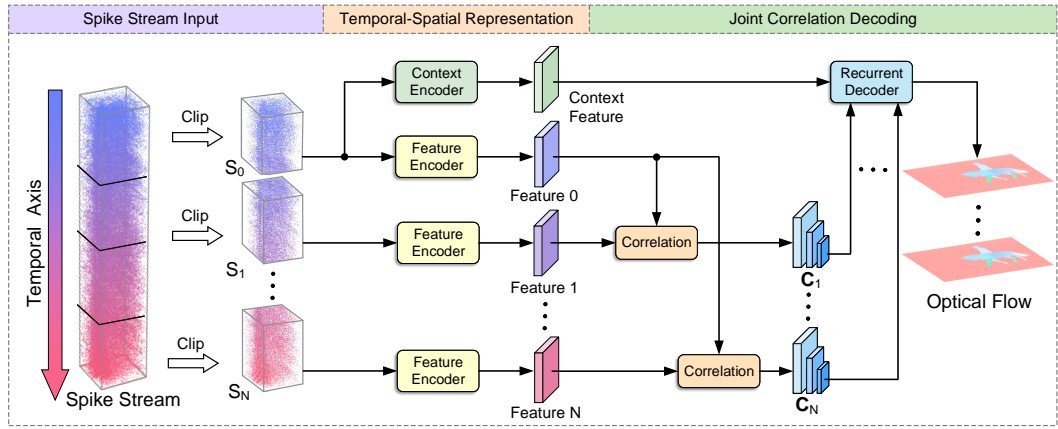

Figure 1: The overall architecture of the Spike2Flow. The input spike stream is firstly clipped to several sub-streams. Each sub-stream is extracted to be a feature for correlation through temporal-spatial representation, and the first feature constructs correlations with all other features. The recurrent decoder estimates flow fields from all the correlations and the context feature.

where $\rightarrow$ means pixel-level registration. Given a spike stream $S$, the target of optical flow estimation is estimating $\mathbf{w}$. The data format of the spike stream is continuous binary matrices, which is different from the classic data pattern of video. Correspondingly, the processing method should also be different. Firstly, unlike a pixel in videos, a single spike represents the result of the integral procedure rather than the status at the current moment, and a single spike cannot express spatial-temporal information. Secondly, the spike streams are more continuous than classic videos. Thus, taking advantage of the continuousness for analyzing the scene is a challenge and an opportunity.

**Data Generation.** SCFlow proposed two synthetic datasets through a graphics simulator for training and evaluation, respectively. However, there are several limitations of the simulating and rendering, such as unrealistic textures, overly simplified lighting conditions, and unreasonable appearance. The huge gap between the synthetic scenes and the real world causes the models often have poor generalization on the real domain due to the gap [34; 45; 63; 7]. Thus, it is essential to train networks with data from real scenes. To improve the generalization on the real domain, we use a high-speed dataset Slow Flow [25] with a high spatial and temporal resolution to synthesize an optical flow dataset, i.e., real scenes with spike and flow (RSSF) for spike cameras with real scenes. We use the raw data of Slow Flow to generate RSSF. The raw data have 41 scenes and sum to tens of thousands of frames, and there are three kinds of spatial resolution: $2560 \times 2048, 2560 \times 1440$, and $2048 \times 1152$. We select 11 scenes to generate the testing set and the other 30 scenes to generate the training set. More details are included in supplementary materials. The generation pipeline is as follows.

Firstly, we imitate the image signal processor (ISP) to process the raw data in Bayer pattern to color images, where the operations include demosaicing, white balance, and intensity mapping. The image frames are $2\times$ downsampled spatially and temporally for saving computing and storage sources. Secondly, we use GMA [27] network trained on data mixed by Sintel [5], KITTI [42], HD1K [31] and FlyingThings [40] to generate the reference optical flow of the image frames. Thirdly, We use the reference flow to simulate the motion of the scene and construct a virtual spike camera to fire the spike streams. We interpolate 20 time steps of images between every two adjacent frames for generating spikes, and we use 10 times temporal oversampling to improve the precision of the scene. The ground truth contains flow for the duration of $20, 40$, and $60$ interpolated images, denoted as $dt = 20, dt = 40$, and $dt = 60$, respectively. It is noted that although the reference flow is not absolutely accurate, it is still reliable due to the following reasons.

(1) **High-quality data.** The images for generating the spike have a high spatial and temporal resolution, which can approximate the ideal high-speed scenes that we can hardly get in the real world. Estimating the reference flow from such high-quality data is reliable.

(2) **Correspondence between spike and reference flow.** Although the reference flow fields are not absolutely accurate for color frames, they correspond to the spike streams since the spike streams are generated from the reference motion. Thus, the reference flow fields are reliable for the spike streams.

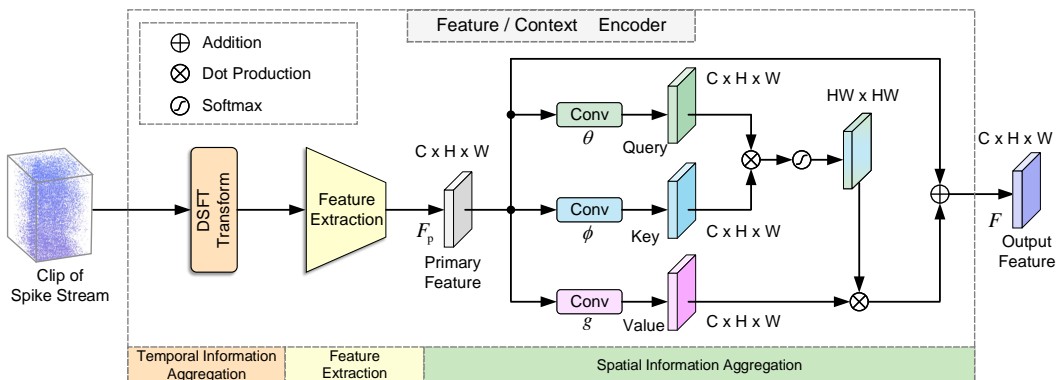

Figure 2: Illustration of temporal-spatial representation (TSR) for spike streams. A spike sub-stream is firstly aggregated temporal information through the differential of spike firing time (DSFT) transform. A primary feature is then extracted from the DSFT to the primary feature with downsampling. Then the primary feature is processed by the spatial information aggregation (SIA) module.

(3) **Good generalization.** Results in section 4.2 show satisfactory performance of our network on PHM dataset [17] and real data. For PHM, the ground truth flow is nearly absolutely reliable, which indicates the generalization of synthetic data and the reliability of our training data. Good performance on real data demonstrates the generalization of our training data on real scenes although there is a domain gap between real and simulated spikes.

### 3.3 Overall Architecture of the network

The overall architecture is shown in Fig. 1. The input spike stream $S$ is firstly clipped to spike sub-streams $\{S_0, S_1, \ldots, S_N\}$. Each sub-stream is extracted to be the feature for matching $\{F_0^M, F_1^M, \ldots, F_N^M\}$, which represents the central moment of the corresponding spike sub-stream, and the first feature $S_0$ is extracted to be context feature $F_0^C$ like RAFT [54]. Noted that it is hard to extract rich features directly from binary spike stream, we propose temporal-spatial representation (TSR) for the above-mentioned feature extraction. To utilize the continuousness of the spike streams, the first matching feature $F_0^M$ constructs all-pairs correlations $\{C_1, \ldots, C_N\}$ with all other matching features. The recurrent decoder jointly estimates flow fields from the central time of $S_0$ to the central time of $\{S_1, \ldots, S_N\}$ through all the correlations and the context feature $F_0^C$.

For better embedding the spike streams to the feature domain, we propose to aggregate the information in the temporal and spatial domains. We design the differential of spike firing time (DSFT) in the temporal domain to transform the spike stream from the binary domain, which aggregates information along the temporal axis for each pixel. Based on DSFT, the spike stream is then embedded in a high-dimensional feature with a downsampling operation. A spatial information aggregation (SIA) module is proposed to extract the information in the spike further. To utilize the continuousness of spike streams, which reflects the procedure of motion, we propose a joint correlation decoding (JCD) module to jointly estimate a series of flow fields starting from the same moment.

### 3.4 Temporal-Spatial Representation for Spikes

**Differential of Spike Firing Time.** Each spike in spike streams represents the result of the integral procedure of photons rather than status at the current moment. Different "1" correspond to various light intensities since the "1" in the spike streams represents the number of accumulated photons rather than the arrival rate of the photons. Using features extracted from the binary spikes for matching may not be appropriate to reflect the structures of scenes. We propose to represent the information contained in the spike through the firing time, which can better represent the arrival rate of the photons at each pixel, i.e., the light intensity at each pixel. For better aggregating the temporal information, as shown in Fig. 3, we propose to use the differential of spike firing time to represent the binary spike. If we denote the DSFT of spike stream $S = S(\mathbf{x}, t)$ as $D = D(\mathbf{x}, t)$, the DSFT transform can be

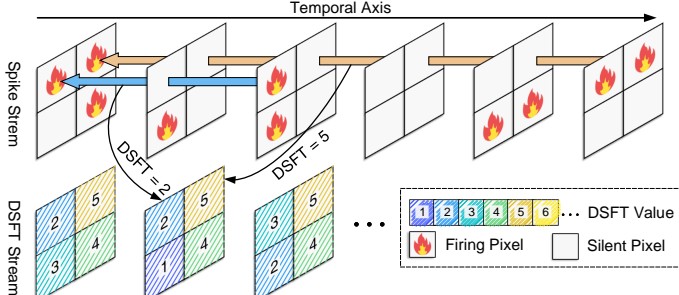

Figure 3: Illustration of the differential of spike firing time (DSFT) Transform. Each pixel in the binary spike stream (the first row in the figure) is represented as the difference in firing time (the second row in the figure) of the corresponding pixel.

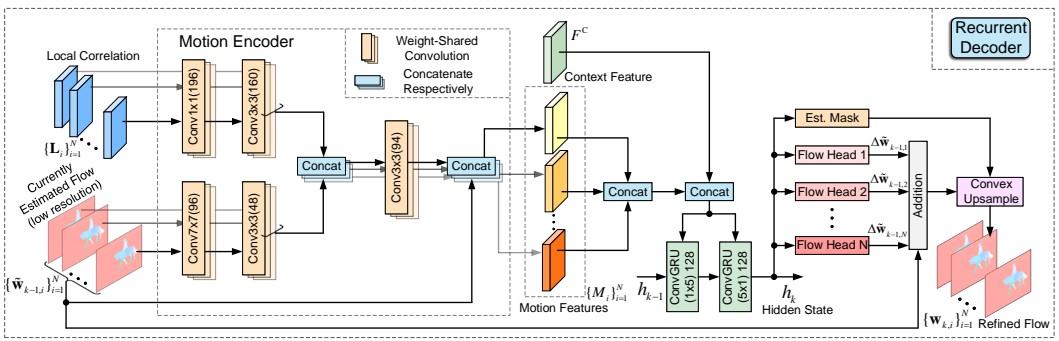

Figure 4: Illustration of the detailed structure of the recurrent decoder. Local correlations are looked up from the correlation and encoded to be motion features with the corresponding flow fields. The context feature and all the motion features are used to update the hidden state of the ConvGRUs. A series of flow fields are decoded with different prediction heads.

formulated as:

$$
\begin{aligned}
D(\mathbf{x}, t) = \mathcal{D}[S(\mathbf{x}, t)] &= T_{\text{next}}(\mathbf{x}, t) - T_{\text{pre}}(\mathbf{x}, t) \\
&= \{\min(\tau) \mid S(\mathbf{p}, \tau) = 1, \ \tau > t\}_{\mathbf{p} \in \mathbb{D}(\mathbf{x})} - \{\max(\tau) \mid S(\mathbf{p}, \tau) = 1, \ \tau \leq t\}_{\mathbf{p} \in \mathbb{D}(\mathbf{x})}
\end{aligned}
\tag{3}
$$

Where $\mathcal{D}$ is the DSFT transform. $T_{\text{next}}(\mathbf{x}, t)$ and $T_{\text{pre}}(\mathbf{x}, t)$ denote the firing time of the next and previous spike at spatial-temporal moment $(\mathbf{x}, t)$. $\mathbb{D}(\mathbf{x})$ denote the domain of definition of $\mathbf{x}$. We use DSFT to reflect the firing rate of the spikes. The firing rate here is a statistical concept, and the average of individual spike intervals is connected with the firing rate. The motion in the scenes changes the brightness, and the Poisson process of photon arrivals causes fluctuations in the firing rate. Thus, the firing rate is temporally variational at each pixel. DSFT can better recover the spikes' dynamic process than obtaining a more constant firing rate with longer time windows.

**Spatial Information Aggregation.** A single spike in space-time coordinates can hardly describe the scene, and we need a group of spikes to represent each pixel. The DSFT aggregates the information in the temporal domain. However, the DSFT is still fluctuating since the arrival of the photons follows the Poisson process, and the firing rate of spikes exhibits remarkable randomness. The fluctuation of spikes can make the value of features for matching unstable. To enhance the features of spike for better matching, we propose a spatial information aggregation (SIA) module to integrate context information with a larger receptive field. As shown on the right side of Fig. 2, we use the self-attention mechanism to aggregate the spatial information, which can be formulated by:

$$
F = F_{\text{p}} + \text{softmax}\left(\theta(F_{\text{p}}) \, \phi(F_{\text{p}})^T\right) \cdot g(F_{\text{p}})
\tag{4}
$$

Where $F$ denotes the feature finally output by the encoder. $F_{\text{p}}$ denotes the primary feature before SIA. $\theta, \phi$, and $g$ embed the primary feature to query, key, and value, respectively. Through the SIA module, each pixel gets a long-range aggregation with pixels in its relative area.

## 3.5 Joint Decoding of Correlation

There is more continuously temporal information in spike streams compared with videos. To extract the continuous moving procedure of the optical scene in spike, we propose to jointly decode a series

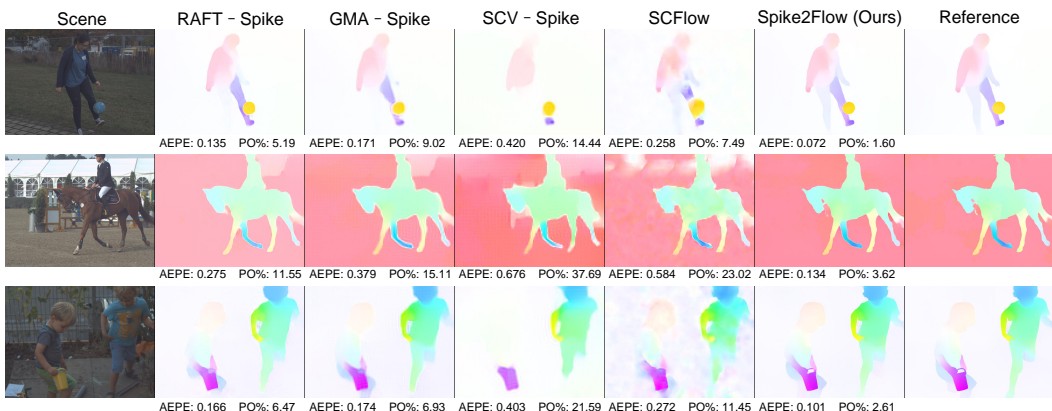

Figure 5: Visual results on RSSF in $dt = 20$ case, the meaning of each column is on the top. PO% means the percentage of outliers. **Please enlarge the figures for better comparison**.

of motions from the same starting moment, i.e. $\{\mathbf{w}(\mathbf{x}, t \mid t_0)\}$. To simplify the computation, we sparsely sample the motion and estimate $\{\mathbf{w}_i = \mathbf{w}(\mathbf{x}, iT + t_0 \mid t_0)\}_{i=1}^N$, where $T$ is the sampling cycle, $i$ is the sampling index, and $N$ is the maximum sampling index. For each sampling moment, we construct a multi-scale all-pairs correlation [54] from the matching features $F_0^M$ and $F_i^M$:

$$\mathbf{C}_i^l(x, y, m, n) = \frac{1}{2^{2l}} \sum_{p=0}^{2^l - 1} \sum_{q=0}^{2^l - 1} \left\langle F_0^M(x, y), F_i^M(m \cdot 2^l + p, n \cdot 2^l + q) \right\rangle \tag{5}$$

Local correlations $\{\mathbf{L}_1, \ldots, \mathbf{L}_N\}$ are then looked up through current estimated flow fields with a radius $r$. The local grid for looking up can be formulated as:

$$\mathcal{N}(\mathbf{x})_{i,r,l} = \{(\mathbf{x} + \mathbf{w}(\mathbf{x}, iT + t_0 \mid t_0))/2^l + \mathbf{y} \mid \mathbf{y} \in \mathbb{Z}^2, \|\mathbf{y}\|_1 \le r\} \tag{6}$$

The structure of the recurrent decoder is shown in Fig. 4. In each iteration, the decoder optimizes the flow fields by estimating their residual through local correlations and context feature $F_0^C$. Suppose that the start moment of the flow is $t_s$, and the sampling cycle of the flow is $T$. The local correlations $\{\mathbf{L}_i\}_{i=1}^N$ and currently estiamted flow fields $\{\mathbf{w}_{k-1,i} = \mathbf{w}_{k-1}(t_s + iT \mid t_s)\}_{i=1}^N$ are extracted to be motion features $\{M_i\}_{i=1}^N$ by the motion encoder, where $k$ is the iteration index of the ConvGRUs. All the motion features $\{M_1, \ldots, M_N\}$ and $F_0^C$ are concatenated and input to the ConvGRUs [8] to update the hidden states. For all the flow fields to be estimated, we use a single hidden state. Different prediction heads are employed to estimate the residual of different flow fields.

Suppose the recurrent decoder has $V$ iterations. Given ground truth flow fields $\{\mathbf{w}_i^{gt}\}_{i=1}^N$ the loss function is defined as:

$$\mathcal{L} = \sum_{i=1}^N \sum_{j=1}^V \gamma^{V-j} \|\mathbf{w}_i^{gt} - \mathbf{w}_{i,j}\|_1 \tag{7}$$

where $\gamma$ is the decay factor set as $0.8$.

## 4 Experimental Results

### 4.1 Implementation Details

In the experiments, we set $N$ as 3 and T as 20, which means we jointly estimate optical flow under $20, 40$, and $60$ time steps difference. We set the number of input spike frames as 21. For constructing correlation, we set the multi-scale level as 3, and we set the looking-up radius $r = 3$. The model is trained on the training set of the real scenes with the spike and flow (RSSF) dataset. We randomly crop the spike stream to $320 \times 448$ spatially during the training procedure and set the batch size as 6. We use randomly horizontal and vertical flips as data augmentation to balance the motion in the

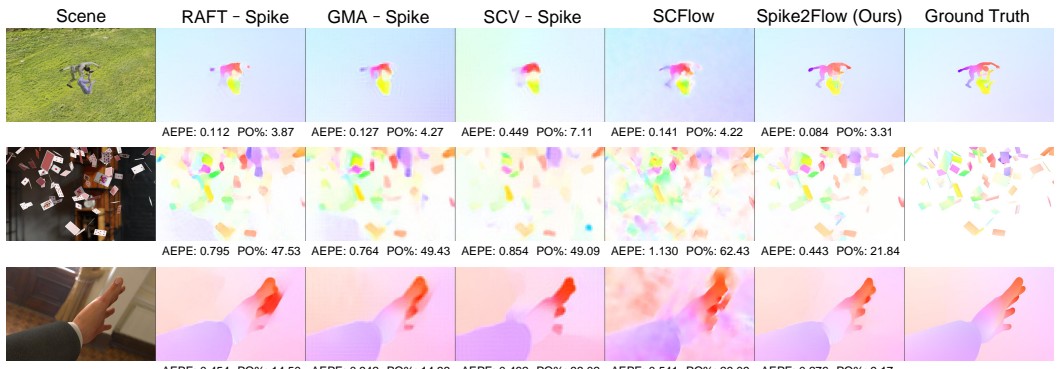

| Scene | RAFT - Spike | GMA - Spike | SCV - Spike | SCFlow | Spike2Flow (Ours) | Ground Truth |
|---|---|---|---|---|---|---|
| | AEPE: 0.112 PO%: 3.87 | AEPE: 0.127 PO%: 4.27 | AEPE: 0.449 PO%: 7.11 | AEPE: 0.141 PO%: 4.22 | AEPE: 0.084 PO%: 3.31 | |
| | AEPE: 0.795 PO%: 47.53 | AEPE: 0.764 PO%: 49.43 | AEPE: 0.854 PO%: 49.09 | AEPE: 1.130 PO%: 62.43 | AEPE: 0.443 PO%: 21.84 | |
| | AEPE: 0.454 PO%: 14.50 | AEPE: 0.342 PO%: 14.33 | AEPE: 0.462 PO%: 33.02 | AEPE: 0.541 PO%: 29.09 | AEPE: 0.276 PO%: 9.17 | |

Figure 6: Visual results on PHM in $dt = 10$ case, the meaning of each column is on the top.

Table 1: Quantitative results on the testing set of RSSF. All the models are **retrained** on the training set of RSSF based on the official code. The best results over each group are bolded. PO%: the percentage of outliers.

| Method | $dt = 20$ | | $dt = 40$ | | $dt = 60$ | |
|---|---|---|---|---|---|---|
| | AEPE | PO% | AEPE | PO% | AEPE | PO% |
| RAFT [54] – AvgImg | 0.244 | 8.08 | 0.390 | 12.41 | 0.573 | 14.85 |
| RAFT [54] – Spike | 0.181 | 5.62 | 0.295 | 8.69 | 0.437 | 10.71 |
| SCV [28] – AvgImg | 0.534 | 33.86 | 0.762 | 39.54 | 1.056 | 42.29 |
| SCV [28] – Spike | 0.570 | 40.03 | 0.808 | 43.90 | 1.132 | 44.34 |
| GMA [27] – AvgImg | 0.226 | 7.20 | 0.388 | 12.54 | 0.632 | 14.81 |
| GMA [27] – Spike | 0.230 | 7.45 | 0.401 | 11.60 | 0.603 | 14.16 |
| SCFlow [17] | 0.389 | 14.00 | 0.668 | 19.00 | 1.264 | 23.40 |
| Spike2Flow(ours) | **0.117** | **2.57** | **0.197** | **5.18** | **0.286** | **6.98** |

training dataset. We use Adam optimizer [30] with $\beta_1 = 0.9$ and $\beta_2 = 0.999$. The learning rate is initially set as 3e-4 and scaled by 0.7 every 10 epoch. The model is trained for 100 epochs.

We use average end-point error (AEPE) and percentage of outliers (PO%) as evaluating metrics. AEPE is the spatial average of Euclidean distance between optical flow $\mathbf{w}$ and the ground truth $\mathbf{w}_{\mathrm{gt}}$:

$$\mathrm{AEPE} = \frac{1}{HW} \sum_{\mathbf{x}} \|\mathbf{w}(\mathbf{x}) - \mathbf{w}_{\mathrm{gt}}(\mathbf{x})\|_2 \tag{8}$$

where $H$ and $W$ are the height and width of the flow. The percentage of outliers is the percentage of the pixel with end-point error larger than **0.5** and **5%** of its ground truth at the same time.

## 4.2 Comparative Results

We compare our method with SCFlow [17] and methods straightforwardly designed based on RAFT [54], SCV [28], and GMA [27] for estimating optical flow for spike cameras. There are two ways to input the spikes to the methods for optical flow estimation for spike cameras:

(1) **AvgImg**. Averaging the spike stream along the temporal axis as a gray image.

(2) **Spike**. Directly input the spike streams into the network work as a multi-channel image.

We do not use the event-based optical flow architecture since it has been proven to be noneffective in [17], and we do not use the training set in [17] since it is synthetic and the generalization ability is limited. We use "RAFT – AvgImg" and "RAFT – Spike" to represent the direct adaptation based on RAFT with AvgImg and spike, respectively. Adapted methods for other architectures are the same. All the methods are **retrained** under settings in section 4.1 based on the official code on RSSF training set with spike streams as input, respectively. Noted that Spike2Flow jointly estimates flow fields with time duration $dt = 20, 40, 60$. We set the training data for comparable models and

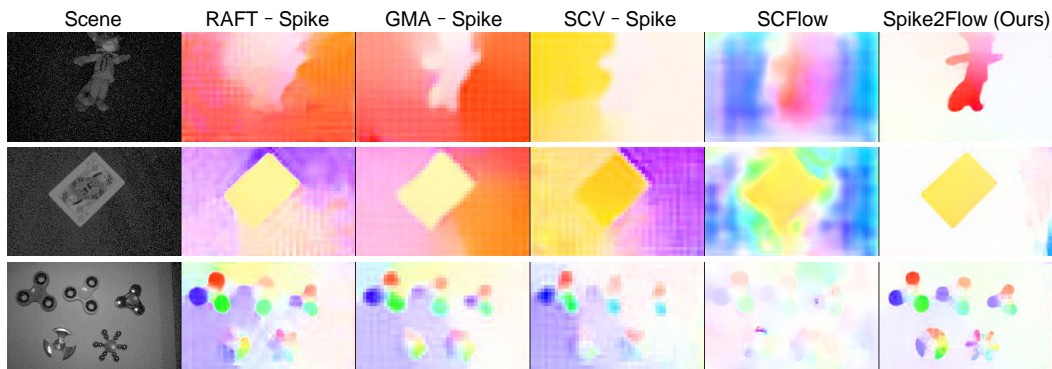

| Scene | RAFT – Spike | GMA – Spike | SCV – Spike | SCFlow | Spike2Flow (Ours) |

Figure 7: Visual results on real data in $dt = 20$ case, the "Scene" is the average of spike stream on temporal axis with gamma transform.

Table 2: Quantitative results on PHM. All the models are **retrained** on the training set of RSSF based on the official code. The best results for each group are bolded. PO%: the percentage of outliers.

| Method | $dt = 10$ | | $dt = 20$ | |
|---|---|---|---|---|
| | AEPE | PO% | AEPE | PO% |
| RAFT [54] – AvgImg | 1.635 | 32.55 | 2.753 | 40.85 |
| RAFT [54] – Spike | 0.622 | 21.50 | 1.219 | 23.79 |
| SCV [28] – AvgImg | 1.055 | 44.17 | 1.829 | 54.24 |
| SCV [28] – Spike | 0.841 | 34.18 | 1.466 | 41.26 |
| GMA [27] – AvgImg | 0.874 | 28.30 | 1.784 | 39.67 |
| GMA [27] – Spike | 1.485 | 27.41 | 1.888 | 29.25 |
| SCFlow [17] | 1.027 | 34.54 | 1.775 | 38.57 |
| Spike2Flow(ours) | **0.545** | **16.07** | **1.088** | **19.61** |

randomly select $dt$ from $\{20, 40, 60\}$ for each sample in a mini-batch for the sake of fairness. We evaluate the models on three kinds of data: (1) The testing set of real scenes with spike and flow (RSSF), (2) photo-realistic high-speed motion (PHM), and (3) real data captured by spike cameras.

**Results on real scenes with spike and flow (RSSF)**. The quantitative results on RSSF is shown in Tab. 1. The Spike2Flow jointly estimates the flow in $dt = 20, 40, 60$ cases while other models estimate respectively. Spike2Flow gets the best performance in both AEPE and PO% in all cases. The visual results on RSSF are shown in Fig. 5. All the optical flow color-coding rule is the same with [1]. More results are included in the supplementary material.

**Results on photo-realistic high-speed motion (PHM)**. For comparative experiments on PHM, we use the same model as experiments on the testing set of RSSF, which is trained on the training set of RSSF. The ground truth of optical flow in PHM is generated from a graphics simulator, which is highly reliable. PHM contains ground truth flow in $dt = 10$ and 20 cases. For Spike2Flow, we jointly estimate flow fields in $dt = 10, 20, 30$ cases. For other models, we estimate flow fields in $dt = 10$ and $dt = 20$ cases, respectively. Noted that we select the 9 scenes in PHM except fly since the motion speed of fly is excessive and unrealistic. Quantitative results are shown in Tab. 2, and the results are average among all the scenes we use of the PHM. Spike2Flow still achieves the best performance in all the cases, which demonstrates the good generalization of our method. Compare Tab. 2 with Tab. 1, the performance results on PHM are lower than those on RSSF since the scene in PHM is simulated through a graphics model, and the speed motion of the scenes is ultra-high, which is more challenging for optical flow estimation. The visual results on PHM are shown in Fig. 6 and more results are included in the supplementary material.

**Results on real data**. We evaluate the models on real data captured by spike cameras. The visual results are shown in Fig 7. Although we do not have the ground truth of optical flow in real data, we can see that results of Spike2Flow have sharper edges and more clear background, which shows a better generalization. More details are included in the supplementary material.

## 4.3 Ablation Study

To verify the efficiency of the modules we propose, we implement a series of ablation studies. The quantitative results are shown in Tab. 3. We use color gradation from yellow to green for each column, where greener represents better performance. The results demonstrate the effectiveness of DSFT, SIA, and JCD modules. Although the Exp. (D) and (E) have similar performance in $dt = 20$ and $dt = 40$, Exp. (E) performs better than (D) in the percentage of outliers under $dt = 60$, which shows the effectiveness of the SIA module. Comparison between Exp. (B) and (C) can also show that SIA is effective. Besides proposed modules, we also perform ablation studies on the number of input spike frames (NISF). We select $\{1, 5, 11, 15, 21\}$ as the NISF candidates. The quantitative results are shown in Tab. 4, where we can see larger NISF makes the performance better.

Table 3: Ablation study of our proposed modules. Greener blocks represent better performance with lower AEPE or the percentage of outliers (PO%).

| Index | Setting of experiment | $dt = 20$ | | $dt = 40$ | | $dt = 60$ | |
|---|---|---|---|---|---|---|---|
| | | AEPE | PO% | AEPE | PO% | AEPE | PO% |
| (A) | Removing DSFT transform | 0.236 | 7.69 | 0.382 | 13.49 | 0.536 | 16.94 |
| (B) | Removing SIA and JCD module | 0.128 | 3.30 | 0.218 | 6.37 | 0.325 | 8.40 |
| (C) | Removing JCD module | 0.126 | 3.13 | 0.210 | 5.96 | 0.310 | 7.88 |
| (D) | Removing SIA module | 0.116 | 2.54 | 0.199 | 5.16 | 0.298 | 7.24 |
| (E) | Our final model | 0.117 | 2.57 | 0.197 | 5.18 | 0.286 | 6.98 |

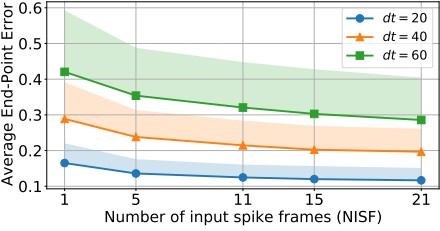

Figure 8: AEPE of ablation on NISF, shadow areas represent the average of the standard deviation of each scene.

Table 4: Ablation study of the number of input spike frames (NISF). Greener blocks represent better performance with lower AEPE or PO%.

| NISF | $dt = 20$ | | $dt = 40$ | | $dt = 60$ | |
|---|---|---|---|---|---|---|
| | AEPE | PO% | AEPE | PO% | AEPE | PO% |
| 1 | 0.174 | 4.37 | 0.315 | 10.85 | 0.463 | 15.01 |
| 5 | 0.153 | 3.44 | 0.277 | 7.70 | 0.417 | 11.35 |
| 11 | 0.125 | 2.98 | 0.218 | 6.30 | 0.323 | 8.55 |
| 15 | 0.123 | 3.01 | 0.211 | 5.83 | 0.312 | 7.85 |
| 21 | 0.117 | 2.57 | 0.197 | 5.18 | 0.286 | 6.98 |

## 5 Conclusions

We propose a method for learning optical flow from continuous spike streams. To improve the generalization of spike-based optical flow, we propose a flow dataset RSSF with real scenes. For better extracting information from binary spike streams, we propose DSFT and SIA modules to extract the temporal and spatial information, respectively. To use the continuousness of spike streams as motion clues, we propose a JCD module to jointly estimate a series of flow fields. Experimental results show that Spike2Flow achieves state-of-the-art performance in spike-based optical flow on RSSF, PHM, and real data, and ablation studies verify the effectiveness of our proposed modules.

**Limitations.** The characteristics of spike firing in extremely dark or bright scenes are quite different, so the same representation strategy may not work well all the time. We plan to extend our model to handle these questions in future work.

## Acknowledgments and Disclosure of Funding

This work is supported by the National Natural Science Foundation of China under Grant 22127807, 62072009, 61931014, and 61972115, and also supported by National Key R&D Program of China under Grant 2021YFF0900501.

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
