# OpenReview forum: "Learning Optical Flow from Continuous Spike Streams"
_NeurIPS.cc/2022/Conference — NeurIPS 2022 Accept_

### Official Review · Reviewer_PUft · 2022-07-10

**Rating:** 4
**Confidence:** 3
**Soundness:** 2 fair
**Presentation:** 1 poor
**Contribution:** 2 fair

**Summary:**

This paper proposes a method, Spike2Flow, to estimate optical flow from videos recorded by spiking cameras. The input modality is rarely used and not properly introduced, so it hard for me to evaluate the merits of this work. Moreover, the technical contribution seems marginal.

**Questions:**

1. Please give illustrations of the output (consecutive frames) of spiking cameras, so that readers will have an intuitive understanding what the videos look like, and what challenges come with this new modality.
2. Why spiking cameras are called with different names in different papers? The two cited references are [11; 19], but in [11] it's called a "spike camera" (instead of spiking), and in [19] it's called "vidar".


**Strengths And Weaknesses:**

Strengths:
1. The empirical performance seems good.

Weaknesses:
1. The presentation is quite unclear. Esp., as spiking cameras are an emerging device and rarely seen, most people are not familiar with its properties, advantages and specific challenges. The authors did not do well when introducing the background of this task.
2. The architecture presented in Fig. 1 seems very similar to RAFT. The Spatial Information Aggregation in Fig.2 seems very similar to CRAFT [a]. It gives me a feeling that the proposed method has rather limited novelty.

[a] CRAFT: cross-attentional flow transformers for robust optical flow. CVPR 2022.

---

> ### Author Response · Authors · 2022-08-02
> **Responses to Reviewer PUft (Part 3/3)**
>
> **Answers to the Questions**
>
> > 1. Please give illustrations of the output (consecutive frames) of spiking cameras, so that readers will have an intuitive understanding what the videos look like, and what challenges come with this new modality.
>
> Thank you for your insightful question. We had included the video illustration of the output of the spiking cameras **in the video of the supplementary material** when first submitting the paper. The video has several examples of comparison of color-coded optical flow. The top-left corner of each example is the dynamic visualization of the binary output of the spiking camera. The 0:10 - 0:26 of the video are about real data captured by spiking cameras. Note that due to downsampling and compression, the visual effect of the binary spikes in the video may seem a little different compared with binary spike frames in original resolution.
>
> We don't include the binary spikes in the body of the paper since they are more appropriate to be seen dynamically. We hope the supplementary material can help the readers better understand the camera. As mentioned in the first question and shown in the top-left corner of the video, the binary spike is non-uniform. Thus, extracting stable features from the binary spike is an important topic.
>
>
>
> > 2. Why spiking cameras are called with different names in different papers? The two cited references are [11; 19], but in [11] it's called a "spike camera" (instead of spiking), and in [19] it's called "vidar".
>
> Thank you for your careful observation and question. The "spike camera", "spiking camera" and "vidar" refer the same camera model that is studied in this paper. The "spike" in "spike camera" is a noun, and it means the data modality of the camera is "spike". The "spiking" in "spiking camera" is a gerund of the verb "spike", and it means the camera represents data by "spiking". Thus, the meaning of these two words are similar.
>
> There are similar cases that the same concept has non-uniform names in emerging areas. For example, "spiking neural networks" [a] is called "spike neural networks" in [b].
>
> [a] W. Fang, et al. Deep Residual Learning in Spiking Neural Networks. NeurIPS 2021.
> [b] L. Zhang, et al. TDSNN: From Deep Neural Networks to Deep Spike Neural Networks with Temporal-Coding. AAAI 2019.

---

> ### Author Response · Authors · 2022-08-02
> **Responses to Reviewer PUft (Part 2/3)**
>
> **Answers to the weaknesses**
>
>
> > 2. The architecture presented in Fig. 1 seems very similar to RAFT. The Spatial Information Aggregation in Fig.2 seems very similar to CRAFT [a]. It gives me a feeling that the proposed method has rather limited novelty.
> >
> > [a] CRAFT: cross-attentional flow transformers for robust optical flow. CVPR 2022.
>
> Thank you for your comments.  Our method has some similarities with RAFT since our method is designed based on RAFT architecture. However, the main contributions of our work is about the processing of spike streams. We concentrate on the two key points below to design our network:
> **(A)** How to extract information from binary spike streams
> **(B)** How to use the continuousness of the spike streams
>
> Point (A) is associated with Section 3.4 of our paper. Firstly, light intensity is significant when estimating optical flow. However, the light intensity of pixels cannot be directly mapped from the "0" and "1" in the binary spike streams. Different "1" correspond to various light intensities since the "1" in the spike streams represents the number of accumulated photons rather than the arrival rate of the photons. Thus, It's not appropriate to construct correlation volumes using features extracted directly from binary spikes for matching. We propose DSFT to transform the spikes into the interval domain. The DSFT of spikes can better represent the  arrival rate of the photons at each pixel, i.e., the light intensity at each pixel. Secondly, the arrival rate of photons reflected by spikes is still fluctuating after DSFT transformation since the arrival of the photons follows the Poisson process. We design the SIA module to use the similarities in the features to reduce the influence caused by the randomness of the data.
>
> Point (B) is associated with Section 3.5 of our paper. We simultaneously construct correlation volumes for a series of continuous motions. The local correlations looked up according to current flows are encoded into a single hidden state for jointly decoding the corresponding flow fields (as shown in Fig. 12 of our pdf file in supplementary material). Jointly estimating the continuous motion can constrain each flow with other flow fields and reduce the randomness of the spikes since the movement is continuous.
>
> CRAFT [a] is a newly proposed method in CVPR 2022, and we didn't notice this paper before submitting the paper. As shown in Fig. 2 and Section 3.1 of [a], the Semantic Smoothing Transformer in CRAFT aims at only transforming the second frame in optical flow estimation to have more context information to construct a better correlation volume with Cross-Frame Attention.
>
> We didn't refer to CRAFT when designing Spike2Flow, and there are differences between CRAFT and our method. We design the SIA module aiming at constructing a general feature extraction scheme for spikes, which is different from CRAFT. The spikes and their interval are non-uniform since the arrival of photons follows the Poisson process. We use the self-attention mechanism to alleviate the non-uniformness of spikes using the non-local similarities. The SIA is applied in the extraction of all the features for correlation and context feature. Self-attention [b]  used in SIA is a common module in deep learning architecture. CRAFT only applies SST on the feature of the second frame, and it has a relative positional encoding that we don't have.
>
> [a] X. Sui, et al. CRAFT: cross-attentional flow transformers for robust optical flow. CVPR 2022.
> [b] A. Vaswani, et al. Attention is all you need. NeurIPS 2017.

---

> ### Author Response · Authors · 2022-08-02
> **Responses to Reviewer PUft (Part 1/3)**
>
> Thank you for your helpful comments, summary of our paper and affirmation of the performance. The questions and my answers are as follows.
>
> **Answers to the weaknesses**
>
> > 1. The presentation is quite unclear. Esp., as spiking cameras are an emerging device and rarely seen, most people are not familiar with its properties, advantages and specific challenges. The authors did not do well when introducing the background of this task.
>
> Thank you for your kind reminder. The properties of spiking camera are important for understanding the mechanism of the camera. However, the space is limited in the paper.
>
> - Properties of spiking camera
>   Spiking camera is an emerging kind of sensor that accumulates photons continuously. A spike is fired when the accumulation reaches the threshold at a pixel. The accumulation of photons and the firing of spikes are high-speed and asynchronous at each pixel. As introduced in Section 3.1 of our paper, each pixel of the spiking camera is composed of 3 key components: (a) photon-receptor, (b) integrator, and (c) comparator. The photon-receptor receives photons and converts them to the voltage in the integrator. A spike is fired when the comparator detects that the accumulation in the integrator reaches the firing threshold. At the same time, the accumulation value of the integrator is reset to 0. The details of the 3 components can be found in Fig. 2 of [a].
>
>   Although each pixel of the spiking camera is asynchronous, the reading of spikes is synchronous and controlled by the clock circuits. Thus, the output of the spiking camera is a binary sequence whose shape is $H \times W \times T$, where $H$ and $W$ are the spatial resolution of the camera and $T$ is the number of reading times. The "1" in the output means the accumulator has reached the firing threshold in the reading moment, and vice versa for "0". Currently, the reading rate is up to 40 kHz, which is pretty high. The spatial resolution of spiking camera is 250 x 400. There will be spiking camera with high spatial resolution up to 1000 x 1000 soon.
>
> - Advantages of spiking camera
>
>   As described in the above paragraph, the spiking camera has a very high temporal resolution, which enables the camera to record high-speed scenes clearly. Besides, the spiking camera can record the continuous changing procedure of the scenes. These two advantages make the camera appropriate for optical flow estimation.
>
> - Specific challenges of spiking camera
>
>   (a) Spikes from the spiking camera are in a new kind of data modality. Extracting features from spikes and using the spikes to estimate optical flow are challenges.
>   (b) Compared with traditional image sequences, the spike streams are continuous. Using the information in continuous stream data is also a challenge.
>   The (a) and (b) above are associated with contribution (1) (line 48-51) and contribution (2) (line 52-53) in our paper respectively.
>
> More details of the properties of spiking camera can be found in Section 2 of [a] and [b].
>
> [a] J. Zhao, et al. Reconstructing Clear Image for High-Speed Motion Scene With a Retina-Inspired Spike Camera. TCI 2022.
> [b] T. Huang, et al. 1000x Faster Camera and Machine Vision with Ordinary Devices. Engineering 2022.

---

> > ### Comment · Reviewer_PUft · 2022-08-08
> > **Thanks for the clarification**
> >
> > I raised my rating to borderline reject.
> >
> > However, I still feel a few limitations exist: 1) the problem setting is not fully motivated. What's the advantages of spiking camera? Only higher frame rate? The authors even don't have a real dataset. They merely converted Slow Flow (RGB videos) to spiking videos for evaluation (not to mention the converted videos may have a domain gap with real spiking videos). 2) Moreover, it seems that since the method is an adaptation on the new modality of spiking cameras, maybe it more suits an application-oriented conference like CVPR, instead of NeurIPS, which considers broader impact. 3) I still believe that the technical novelty (i.e., on the optical flow estimation model design) is rather limited. So I'll keep the rating at the negative side.

---

> > > ### Author Response · Authors · 2022-08-08
> > > **Responses to Reviewer PUft (Part 2/2)**
> > >
> > > **2) Topics of NeurIPS.**
> > > We admit that camera and imaging related research works (like this one) may not be the mainstream in the scope of NeurIPS. However, we noticed that there are still some NeurIPS papers on neuromorphic cameras and other novel cameras (including event cameras, spiking cameras and polarization cameras) and related intelligent algorithms in recent years. For example, The work [a] implements high dynamic range imaging with data from **modulo camera**, and they also train the networks with synthetic data. It is notable that the work [a] implements modulo camera based on **spiking camera** that is studied in our paper to get new real data. The work [b] dehazes images with **polarization camera**, and they also train networks with synthetic data. The work [c] focuses on optical flow estimation for **event camera**. The work [d] studies object detection for high-resolution **event camera**. Thus, we think out paper is still relevant to the scope of NeurIPS.
> > >
> > > [a] C. Zhou, et. al. UnModNet: Learning to Unwrap a Modulo Image for High Dynamic Range Imaging. NeurIPS 2020.
> > > [b] C. Zhou, et. al. Learning to dehaze with polarization. NeurIPS 2021.
> > > [c] J. Hagenaars, et. al. Self-Supervised Learning of Event-Based Optical Flow with Spiking Neural Networks. NeurIPS 2021.
> > > [d] E. Perot, et. al. Learning to Detect Objects with a 1 Megapixel Event Camera. NeurIPS 2020.
> > >
> > > **3) The design of the model.**
> > > We design the model based on the challenges and key advantages of spiking camera when estimating optical flow. The contributions are proposed according to the characteristics of the spiking camera.
> > > (1) DSFT estimates the firing rate of spikes to represent the light intensity. The firing rate fluctuates since the photon arrivals follow a Poisson process, and the motion of objects makes them projected on different pixels, making the firing rate change dramatically. We use continuous DSFT rather than the  statistical term of firing rate to recover the dynamic process better.
> > > (2) As described in (1), the firing of spikes exhibits remarkable randomness since the photon arrivals follow the Poisson process. We propose SIA module to reduce the influence of the randomness of spikes in all feature extraction of spikes. After feature extraction, we consider the motion as a continuous dynamic process and construct a series of correlation volumes. The local correlations are encoded to a single hidden state for estimating optical flow of each moment with the reference of correlations of other moments. This JCD module further utilizes the continuousness of the spike streams in a longer time range to restrain the error in the flow estimation and reduce the influence of spikes' randomness.

---

> > > ### Author Response · Authors · 2022-08-08
> > > **Responses to Reviewer PUft (Part 1/2)**
> > >
> > > We sincerely thank you for your raising rating and further discussion. Our answers to your additional questions are as follows.
> > >
> > > **1) - (a) Advantages of spiking camera.**
> > > Spiking camera has other advantages, such as high dynamic range, low latency, and high memory efficiency, which has been introduced in line 18 - 19 of our paper.
> > >
> > > Firstly, the dynamic range depends on the maximum and minimum signal that the device can record. In traditional camera, the maximum signal depends on the capacity of the photon well, and the minimum signal depends on thermal noise and reading noise. For the same device in traditional camera, the dynamic range is proportional to the capacity of the photon well. In spiking camera, the accumulation in the photon well will reset to zero conditionally. In a single reading cycle, the "capacity" of photon well of spiking camera is limited, but it increases with the recording time, i.e., the number of spike frames. Thus, the spiking camera has a **high dynamic range**, and there has been a method to use spiking camera to guide high-dynamic imaging for traditional camera [a].
> > >
> > > Secondly, in traditional cameras, all the pixels have the same exposure time, which is usually set far below the limit of the CMOS device to expose the dark areas and restrain the noise. In spiking camera, there is not a uniform exposure time, and we can set a **much higher frame rate** that is more close to the limit of the CMOS compared with traditional camera. The optical scenes are converted to spike streams at high speed. Thus, the spiking camera has **low latency**.
> > >
> > > Thirdly, compared with traditional camera that has the same frame rate, spiking camera needs **much less memory and bandwidth** since the spike streams are sparse and binary.
> > >
> > > Given the above-mentioned advantages, the spiking camera has broad application scenarios, especially in high-speed scenes. In the applications of spiking camera, estimating optical flow is crucial since it offers clues to the motion in the scene.
> > >
> > > [a] J. Han, et. al. Neuromorphic Camera Guided High Dynamic Range Imaging. CVPR 2020.
> > >
> > > **1) - (b) Real Dataset.**
> > > There are experiments on real data that have been introduced in our answer to Weakness 2 of Reviewer zkLu ([Link](https://openreview.net/forum?id=3vYkhJIty7E&noteId=bmdsOSNV9DW)). As for the generalization of simulated spikes, we have introduced in line 164 - 168 of our paper that our method generalizes well on PHM and real data. Thanks for your suggestion, and we have included more descriptions of the domain gap with real spikes in our revised paper.

---

> > > ### Author Response · Authors · 2022-08-09
> > > **Response to Reviewer PUft**
> > >
> > > Thank you for your precious time and valuable comments. Please let us know if you have any further questions, and we will be glad to discuss with you to provide further details about our work if you like.

---

### Official Review · Reviewer_pFP6 · 2022-07-10

**Rating:** 5
**Confidence:** 4
**Soundness:** 3 good
**Presentation:** 3 good
**Contribution:** 2 fair

**Summary:**

The authors present an algorithm for optical flow from spiking camera data. The main contributions are:

1) the introduction of a spike based optical flow network, Spike2Flow which extracts features from binary spikes and estimates flow fields

2) a dataset for spike-based optical flow

3) Experimental results demonstrating good comparative results on various datasets and compared to other algorithms

**Questions:**

See my comments above

**Limitations:**

N/A: not applicable

**Strengths And Weaknesses:**

Strengths:

S1: the paper is well written overall

S2: the paper deals with an interesting problem, namely using spike cameras for optical flow. Given that spike cameras have high temporal resolution, optical flow is a good application for these cameras

S3: the authors demonstrate good results compared to other similar algorithms

S4: the authors indicate that source code will be released upon publication

Weaknesses:

W1: the algorithm does not really seam terribly innovative. This made me a bit ambivalent in deciding the paper's final ranking, but given the good results, I think it should be of interest to the community that deals with spike and event cameras

W2: Eq 1: clarify it there is at most 1 spike per time intervale T=[T_pre, t]. Do all the spikes emerge from all pixels at the same time of is this asynchronous

W3: How much data is actually produces by this sensor per second? The potential issue I see here is that neuromorphic chips that process event data asynchronously typically have I/O limitations. It is not clear how suitable this algorithm would be for such hardware. Furthermore the use of floating point operations (see softmax in eq 4 for example) may limit the applicability of this algorithm in power efficient hardware. A comment on this would be good.

W4: line 209: typo: "sparsely sampling"

---

> ### Author Response · Authors · 2022-08-02
> **Responses to Reviewer pFP6**
>
> Thank you for your summary of our contributions and strengths. Your comments are helpful to our paper, and my answers are as follows.
>
> > 1. the algorithm does not really seam terribly innovative. This made me a bit ambivalent in deciding the paper's final ranking, but given the good results, I think it should be of interest to the community that deals with spike and event cameras
>
> Thank you for your comments and affirmation that the paper should be of interest to the community of spike and event camera. Our innovations are mainly focused on the challenges of the spiking camera. The key points in this paper to estimate optical flow from spike streams are:
>
> - Extracting stable and efficient features from binary spikes
> - Using the continuousness of the spike streams
>
> The method for feature extraction is in Section 3.4 of our paper. Firstly, light intensity is crucial in optical flow estimation. However, different "1" correspond to various light intensities in spike streams since the "1" reflects the accumulation of the photons at a pixel rather than the photons' arrival rate, i.e., the light  intensity. Using features directly extracted from binary spikes for matching in correlation volumes may be inappropriate. We design DSFT to transform the spikes into the interval domain to represent the light intensities at each pixel better. Secondly, the light intensities reflected by the spikes are still fluctuating since the arrival of the photons follows the Poisson process. We design the SIA module to aggregate long-range relationships for features of the spikes to reduce the randomness and non-uniformness of the spikes with non-local similarities.
>
> The method for using continuousness is in Section 3.5 of our paper. We simultaneously construct a series of correlation volumes for a motion procedure. The local correlations looked up according to current flows are encoded to a single hidden state for decoding the motion procedure. The detailed illustration is shown in Fig. 12 of the pdf file in the supplementary material. In this way, we can use the continuousness of the motion to constrain each flow field using all the other motion clues. Besides, jointly using a series of correlations can also reduce the influence of the randomness in the spikes.
>
>
>
> > 2. Eq 1: clarify it there is at most 1 spike per time intervale T=[T_pre, t]. Do all the spikes emerge from all pixels at the same time of is this asynchronous
>
> Thank you for your insightful comments. In the current spiking camera model, there is indeed at most 1 spike per time interval, and this is a detail about the working mechanism of the spiking camera. There is a firing threshold $\theta$ in Eq. 1 to control the firing of spikes. Actually, the $\theta$ is adjustable in spiking camera, and we adjust the $\theta$ to ensure there is no more than 1 spike is fired during a reading interval. For each pixel of the camera, the spike can be fired at an arbitrary time, and the firing is asynchronous. The reading out of the spikes for all the pixels is synchronous, and it's at a pretty high rate.
>
>
>
> > 3. How much data is actually produces by this sensor per second? The potential issue I see here is that neuromorphic chips that process event data asynchronously typically have I/O limitations. It is not clear how suitable this algorithm would be for such hardware. Furthermore the use of floating point operations (see softmax in eq 4 for example) may limit the applicability of this algorithm in power efficient hardware. A comment on this would be good.
>
> Thank you for your helpful question. Here are my comments.
>
> The spatial resolution of the current implementation of spiking camera is $250 \times 400$, and it outputs $40{\rm k} = 4 \times 10^4$ binary frames per second. The bandwidth of these data is $250 \times 400 \times 4 \times 10^4 \times 1 \ {\rm bit/s} = 4 \times 10^9 \ {\rm bits/s}$. Thus, the bandwidth of the data output from the camera is $\frac{1}{8} \times 4 \times 10^9 \ {\rm Bytes/s} = 5 \times 10^8 \ {\rm Bytes/s} = 476.83 \ {\rm MB/s}$
>
> Transmitting the data is realizable for the spiking camera with PCIe Interface. So does the spiking camera in the next generation with $1000 \times 1000$ spatial resolution, whose bandwidth is around $4.66 \ {\rm GB/s}$.
>
> Applying the methods to neuromorphic chips is a popular topic in the community of neuromorphic cameras. Currently, we mainly focus on methods based on traditional artificial neural networks in the float domain to handle the challenges in optical flow estimation for spiking camera. In future research, we will consider studying energy-efficient methods such as methods based on the binary spiking neural networks to apply optical flow for spiking camera in power-efficient hardware.
>
>
>
> > 4. line 209: typo: "sparsely sampling"
>
> Thank you for your careful observation. We have revised the "sampling" to "sample" in the paper.

---

> ### Comment · Reviewer_pFP6 · 2022-08-07
> **rebuttal**
>
> the authors have answered my questions. I do not see a reason to change the paper's ranking

---

> > ### Author Response · Authors · 2022-08-09
> > **Response to Reviewer pFP6**
> >
> > Thank you for your precious time and valuable comments. Please let us know if you have any further questions, and we will be glad to discuss with you to provide further details about our work if you like.

---

### Official Review · Reviewer_zkLu · 2022-07-11

**Rating:** 5
**Confidence:** 3
**Soundness:** 3 good
**Presentation:** 3 good
**Contribution:** 2 fair

**Summary:**

The authors propose Spike2Flow, a deep neural network that maps spikes from a spiking camera to optic flow estimates. The method encodes spikes with a differential of spike firing time (DSFT) transform. It achieves results competitive with (better than) the state of the art on a new dataset, real scenes with spikes and flow (RSSF).

**Questions:**

1. Are you the first to apply the DSFT / what other encoding methods have been proposed? Can you make the relation with firing rate clearer? Can you show how sensitive this encoding is to noise?
2. Can you apply your trained network to spiking camera inputs? Does it generalize well to non-RSSF, non-"simulated" data?
3. The performance of Spike2Flow is always best. Can the performance of RAFT etc. be mentioned on normal RGB images for the dataset? This puts the performances more in context. For example, does RAFT lose a lot of performance when applying it to AvgImg?
4. Is j-V in equation 7 correct?
5. The dataset RSSF is mentioned as a contribution, but if I understand correctly, it is an automatic adaptation of SlowFlow, which was made by others. Is this correct? Can it be clarified earlier in the text?

**Strengths And Weaknesses:**

Strengths:
* The network focuses on a new type of spiking camera, potentially allowing for very high-speed optic flow estimation.
* The encoding scheme may be useful for other work using spiking cameras.
* The performance of the algorithm seems competitive.

Weaknesses:
* The encoding scheme seems quite straightforward. A spiking camera accumulates illuminance, which means that a bright pixel will have a higher firing rate. The proposed DSFT basically makes a local estimate of this rate, encoding the time between two subsequent spikes. This seems quite a naive method that may be sensitive to noise. Nothing is mentioned about this.
* As far as I understand, the network is not applied to actual spiking camera inputs, but to a simulation of  such a camera based on a normal (or high-speed) camera stream. The paper would be much strengthened by using actual spiking camera inputs.
* RAFT and other sota methods are only applied on the modified  image inputs (AvgImg, Spike). The performance on normal images from the slowflow data set are not mentioned.

---

> ### Author Response · Authors · 2022-08-02
> **Responses to Reviewer zkLu (Part 3/3)**
>
> **Answers to the Questions**
>
> > 1. (a) Are you the first to apply the DSFT / (b) what other encoding methods have been proposed? (c) Can you make the relation with firing rate clearer? (d) Can you show how sensitive this encoding is to noise?
>
> Thank you for your insightful questions. As shown above, we divide the original question into four sub-question (a) - (d).
>
> (a) Are you the first to apply the DSFT?
> Using the interval of spikes to reflect the light intensity of the scene and the accumulation procedure of the spike is straightforward. There are clues in the previous paper on optimization methods to infer the light intensity through the interval of the spike [a]. To the best of our knowledge, we are the first to use the interval of spikes to extract high-dimensional features in deep learning.
>
> (b) What other encoding methods have been proposed?
> There are different encoding methods for spiking camera. The method in [b] (Section IV - A) accumulates the number of spikes of different time lengths to represent different temporal resolutions. The aggregated spike numbers are concatenated in channel dimension and then processed by 1D convolution. The method in [c] processes spike sub-streams in different time lengths using convolution layers and fusing the features using spatial-adaptive attention.
>
> (c) Can you make the relation with firing rate clearer?
> The firing rate is a population parameter of the spikes, while the DSFT is a sample statistic. We use the sample statistic DSFT to estimate and represent the population parameter firing rate, where the statistical DSFT and the estimated firing rate have a reciprocal relation. The firing rate is for a population of spikes, while the DSFT is for a single spike, which is more fine-grained. Besides, we don't estimate the firing rate explicitly in the method. The information on the scene and the accumulation procedure of spikes are contained in the DSFT.
>
> (d) Can you show how sensitive this encoding is to noise?
> We think it's more appropriate to discuss the sensitivity for the whole method than an encoding scheme. Although DSFT can better represent the information in optical flow estimation for spiking camera, there are still fluctuations and randomness in spikes processed by DSFT. The handling schemes for fluctuations are in the following components of the network, which have been introduced in the second paragraph of my answer to weakness 1.
>
> [a] J. Zhao, et al. Reconstructing Clear Image for High-Speed Motion Scene With a Retina-Inspired Spike Camera. TCI 2022.
> [b] X. Xiang, et al. Learning Super-Resolution Reconstruction for High Temporal Resolution Spike Stream. TCSVT 2022.
> [c] J. Zhao, et al. Spk2ImgNet: Learning To Reconstruct Dynamic Scene From Continuous Spike Stream. CVPR 2021.
>
>
>
> > 2. Can you apply your trained network to spiking camera inputs? Does it generalize well to non-RSSF, non-"simulated" data?
>
> Thank you for your concern. We had applied our trained network to real data when first submitting the paper, and it generalizes well. The detailed locations of the results on real (actual) data have been introduced in my answer to weakness 2.
>
>
>
> > 3. The performance of Spike2Flow is always best. Can the performance of RAFT etc. be mentioned on normal RGB images for the dataset? This puts the performances more in context. For example, does RAFT lose a lot of performance when applying it to AvgImg?
>
> Thank you for your comment. We have realized that the original description of the comparable results in the paper may mislead the readers. We have revised the paper to eliminate the misleading. More explanations are in my answer to weakness 3.
>
>
>
> > 4. Is j-V in equation 7 correct?
>
> Thanks for your careful concern. It's a typo, and it should be $V-j$. The output of the last iteration has the highest weight in loss when training. We have revised this typo in the paper.
>
>
>
> > 5. The dataset RSSF is mentioned as a contribution, but if I understand correctly, it is an automatic adaptation of SlowFlow, which was made by others. Is this correct? Can it be clarified earlier in the text?
>
> We select the raw data of SlowFlow dataset as the starting point to simulate the RSSF dataset. There are only image sequences in the raw data of SlowFlow, while we generate flow fields and spikes in RSSF. We indeed use the scene of SlowFlow to generate RSSF, but it's **not an automatic adaptation** to generate the flow fields and spikes. Simulating the spikes according to the scene is non-trivial. We have clarified our contribution on dataset more detailedly in the introduction of our revised paper.

---

> > ### Comment · Reviewer_zkLu · 2022-08-07
> > **Response**
> >
> > Thanks for your clarifications and apologies for missing the results on real spiking camera data.
> >
> > Re 1c: Firing rate does not have to be a neural population characteristic. Firing rate is also a property of a single neuron, or pixel in your camera. If I understand the camera model well, the firing rate is directly related to the brightness. Your DSFT looks at the time between two subsequent spikes to estimate the firing rate of the single pixel. One could also measure the rate over a time window. Longer time windows would give more accurate results for a constant firing rate, but would lag more. My question was if this could be clarified in the text for future readers.

---

> > > ### Author Response · Authors · 2022-08-07
> > > **Response to Reviewer zkLu**
> > >
> > > Thank you for your response and insightful suggestions for our paper!
> > >
> > > In the context of our paper, firing rate refers to the frequency of spike firing in a pixel. In other words, it indicates the **average** number of spikes fired within a certain period of time, i.e., a time window. This term is also connected with the **average** spike interval within that period. This is a **statistical concept**. In the ideal case, if the events of photon arrivals occur with a stable rate, using the "firing rate" and using the "spike interval" can be equivalent.
> > >
> > > However, in reality, the incoming photons follow a Poisson process. Therefore, the process of photon arrivals and the firing of spikes exhibit remarkable **randomness** so that the interval between each individual spike may fluctuate considerably. The existence of thermal noise in the circuits of the sensor may make the fluctuations even worse. Furthermore, we need to further consider the effect of **motion** on the frequency of spike firing. When the objects move rapidly in the scene, the image projected onto the sensor also moves quickly so that the light intensity of a certain pixel may change dramatically. In this way, inferring a "fire rate" from a time window does not make sense because we may combine the spikes triggered by different light intensities and blur the recorded dynamic light intensity changing process.
> > >
> > > In our scheme, we use DSFT to extract the interval of each spike. We choose to keep all these **individual** spike intervals instead of calculating the **statistical** term of firing rate. In this way, we keep more information and detail so that the following network modules are able to recover the dynamic process better.
> > >
> > > We have revised the corresponding descriptions in line 197 - 201 of our paper.

---

> > > ### Author Response · Authors · 2022-08-09
> > > **Response to Reviewer zkLu**
> > >
> > > Thank you for your precious time and valuable comments. Please let us know if you have any further questions, and we will be glad to discuss with you to provide further details about our work if you like.

---

> ### Author Response · Authors · 2022-08-02
> **Responses to Reviewer zkLu (Part 2/3)**
>
> **Answers to the Weaknesses**
>
> > 2. As far as I understand, the network is not applied to actual spiking camera inputs, but to a simulation of such a camera based on a normal (or high-speed) camera stream. The paper would be much strengthened by using actual spiking camera inputs.
>
> Thank you for your concern about the real (actual) data captured from actual spiking cameras. **We had included the results** based on real data captured by actual spiking cameras **in the main paper, supplementary material (pdf and video)** when first submitting the paper. The table below shows 7 scenes of actual spiking camera inputs.
>
> | Index | Name        | Description                       |
> | :---: | :---------- | :-------------------------------- |
> |  (1)  | Fox - A     | A fox doll is shaked              |
> |  (2)  | Poker       | A poker card is falling           |
> |  (3)  | Spining Top | Several spining tops are rotating |
> |  (4)  | Doll        | A Peking Operal doll  is falling  |
> |  (5)  | Fox - B     | A fox doll is shaked              |
> |  (6)  | Fan         | A fan is rotating at a high speed |
> |  (7)  | Leg         | A leg is shaking                  |
>
> Experiments of results on the actually captured data appear in the following locations.
>
> - Fig. 6 in the main body of the paper shows comparable results on real data. The "scene" column is the temporal average of the spike since we think it's inappropriate to show a single frame of raw binary spikes. There are (1), (2), (3) scenes in the above table.
> - Fig. 15 in the pdf in supplementary material shows comparable results on real data. The meaning of the "scene" column is the same as that in Fig. 6. There are (4), (5), (6), (7) scenes in the above table.
> - The 0:10 - 0:26 of the video in supplementary material shows dynamic results on real data. Here we show the raw spike on the top-left corner of the video since the video is temporally dynamic. There are (1), (2), (7) scenes in the above table.
>
>
>
> > 3. RAFT and other sota methods are only applied on the modified image inputs (AvgImg, Spike). The performance on normal images from the slowflow data set are not mentioned.
>
> Thank you for your helpful comment. Your comment reminds me that my statement may mislead the readers of the paper. We have revised the descriptions of the comparable methods in the paper.
>
> The comparison in the paper is to show which method can achieve the best results in the optical flow estimation for spiking camera. There is an important **premise** that the **input** of all the methods is **spike streams**. The RAFT [a] method is designed for RGB images, while in this paper, the "RAFT - Spike" and "RAFT - AvgImg" are not RAFT but straightforwardly designed methods **for estimating optical flow from spike streams** based on key components of RAFT. So does "GMA/SCV - Spike/AvgImg". Our experiment is to show which method trained on RSSF for estimating optical flow **from spike streams** achieves the best performance. Only Spike2Flow, SCFlow and the methods designed straightforwardly are appropriate.
>
> We realize that the original "Method" and "Input" columns in Tab. 1 and Tab.2 may mislead the readers, and we have removed the "Input" column in the paper. Besides, The "AvgImg" is a pre-processing method for spikes to transform the spikes into a gray-scale image, and the "RAFT - AvgImg" is also a method with spikes input.

---

> ### Author Response · Authors · 2022-08-02
> **Responses to Reviewer zkLu (Part 1/3)**
>
> Thank you for your summary, and we appreciate you for pointing out the strengths of our paper. My clarification and answers for the weaknesses and questions you summarized are as follows.
>
> **Answers to the Weaknesses**
>
> > 1. The encoding scheme seems quite straightforward. A spiking camera accumulates illuminance, which means that a bright pixel will have a higher firing rate. The proposed DSFT basically makes a local estimate of this rate, encoding the time between two subsequent spikes. This seems quite a naive method that may be sensitive to noise. Nothing is mentioned about this.
>
> Thank you for your comments. The DSFT indeed makes a local estimate of the spike firing rate through the spike interval. It's natural and acceptable that the spikes are non-uniform and the DSFT transformations of spikes are fluctuating. The fluctuations and randomness are intrinsic characteristics of spiking cameras' output data since **the arrival of the photons follows the Poisson process**. We aim to represent the light intensity better when designing DSFT. The "1" in spike streams reflects the result of the photon integral but not the photon's arrival rate. Different "1" correspond to various light intensities, and using binary spikes to extract features for matching is inappropriate. We use DSFT associated with the arrival rate of the photons to represent the light intensities at each pixel better.
>
> As for handling randomness and fluctuations, we design Spatial Information Aggregation (SIA) and Joint Correlation Decoding (JCD) modules. The encoder firstly extracts deep features from the DSFT of spikes. The SIA module aggregates long-range correspondence in each feature to reduce the influence caused by the randomness of spikes with non-local similarities. The JCD module jointly estimates a series of continuous motions by encoding the corresponding correlations into a single hidden state. The mutual utilization of multiple can counteract the randomness.

---

### Official Review · Reviewer_p5Ne · 2022-07-25

**Rating:** 7
**Confidence:** 4
**Soundness:** 3 good
**Presentation:** 3 good
**Contribution:** 3 good

**Summary:**

The authors have proposed a tailored network that extracts information from binary spikes with ST representation based on the differential of spike firing time and spatial information aggregation. In addition, their proposed network utilizes continuous motion clues through joint correlation decoding.

**Questions:**

From a technical point of view, there isn't any serious concern regarding the content of this work. I would suggest the authors expand the joint decoding of the correlation section a bit more since it is an important part, but the current status of that section is not very readable to the reviewer. For example, I couldn't get the reasoning behind assigning 0.8 to the decay factor here.

**Limitations:**

Compared to previous works, the limitations and impact of the work are adequately outlined and addressed.

**Strengths And Weaknesses:**

All in all, this is a very well-structured paper. An extensive literature review has been performed to support the research objectives of the authors. I especially like the detailed outlining of the architecture of the network.
The comparative results on real scenes with spike and flow as well as photo-realistic high-speed motion are adequately represented, and this demonstrates one of the strengths of this work.

---

> ### Author Response · Authors · 2022-08-02
> **Responses to Reviewer p5Ne**
>
> Thank you for your comprehensive summary of our paper and its strengths. My answers to your questions are as follows.
>
> > 1. The introduction of the joint correlation decoding part.
>
> There are two key points for the network design. One point is to extract efficient features from spikes, and the other point is to use the continuousness of the spike streams. The joint correlation decoding module is designed based on the consideration of the two points. The Spike2Flow network firstly constructs a series of correlation volumes corresponding to the moving procedure, and local correlations are looked up according to the currently estimated flow in each iteration. The JCD encodes the local correlations into a single hidden state for decoding the continuous flow fields. Jointly estimating the motion can constrain the solving for optical flow and reduce the fluctuations of spikes since similar data can counteract the randomness. More details of the JCD module can be found in Section B.1 and Fig. 12 of the pdf file in the supplementary material.
>
>
>
> >2. Why the decay factor is 0.8?
>
> The iterative optimization architecture is common in current optical flow methods. When constructing the loss function, we give the highest weight to the output of the last iteration since it has passed through the most optimization and it's most reliable. 0.8 is an empirical value inherited from RAFT [a].
>
> [a] Z. Teed and J. Deng. RAFT: Recurrent All-Pairs Field Transforms for Optical Flow. ECCV 2020.

---

> ### Author Response · Authors · 2022-08-09
> **Response to Reviewer p5Ne**
>
> Thank you for your precious time and valuable comments. Please let us know if you have any further questions, and we will be glad to discuss with you to provide further details about our work if you like.

---

### Meta-Review · Area_Chair_dRGj · 2022-08-26

**Recommendation:** Accept
**Confidence:** Less certain

**Metareview:**

This work is focused on the estimation of optical flow from a neuromorphic camera that produces Poisson spiking at each pixel with a rate governed by overall intensity. The authors use local space-time aggregation of spike-time differentials to identify features that are then corresponded via a convGRU decoder.

The reviewers found the application interesting, and noted the good performance of the method. There were however a number of concerns about innovation and novelty of the method. Specifically the aggregating spikes to operate on point process data is a standard approach and the assessment of the spiking source of the data was not analyzed. Regardless of the similarity to past methods, overall the reviewers felt that the strengths of the paper, specifically the combination of methods brought together to solve a unique problem, outweighed the weaknesses. Thus I recommend that this work be accepted.

**Award:**

No

---

### Decision · Program_Chairs · 2022-09-14

Accept